# Hypomorphic *SI* genetic variants are associated with childhood chronic loose stools

Bruno P. Chumpitazi[1]*, Jeffery Lewis[2], Derick Cooper[3], Mauro D'Amato[4], Joel Lim[5], Sandeep Gupta[6], Adrian Miranda[7], Natalie Terry[8], Devendra Mehta[9], Ann Scheimann[10], Molly O'Gorman[11], Neelesh Tipnis[12], Yinka Davies[13], Joel Friedlander[14], Heather Smith[3], Jaya Punati[15], Julie Khlevner[16], Mala Setty[17], Carlo Di Lorenzo[18]

1 Baylor College of Medicine, Houston, TX, United States of America, 2 Children's Center for Digestive Health Care, Atlanta, GA, United States of America, 3 QOL Medical, LLC, Vero Beach, FL, United States of America, 4 School of Biological Sciences, Monash University, Clayton, VIC, Australia, 5 Children's Mercy Hospital, Kansas City, MO, United States of America, 6 Sacramento Pediatric Gastroenterology, Sacramento, CA, United States of America, 7 Children's Hospital of Wisconsin, Milwaukee, WI, United States of America, 8 Children's Hospital of Philadelphia, Philadelphia, PA, United States of America, 9 Arnold Palmer Children's Hospital, Orlando, FL, United States of America, 10 Johns Hopkins University, Baltimore, MD, United States of America, 11 Primary Children's Medical Center, Salt Lake City, UT, United States of America, 12 University of Mississippi Medical Center, Jackson, MS, United States of America, 13 Sacramento Pediatric Gastroenterology, Sacramento, CA, United States of America, 14 Children's Hospital Colorado, Digestive Health Institute, University of Colorado School of Medicine, Aurora, CO, United States of America, 15 Children's Hospital of Los Angeles, Los Angeles, CA, United States of America, 16 Columbia University Medical Center, New York, NY, United States of America, 17 UCSF Benioff Children's Hospital Oakland, Oakland, CA, United States of America, 18 Department of Pediatrics, The Ohio State University, Columbus, OH, United States of America

☯ These authors contributed equally to this work.

* chumpita@bcm.edu

**Data Availability Statement:** All relevant data are within the paper and its Supporting Information files.

## Abstract

### Objective

The *SI* gene encodes the sucrase-isomaltase enzyme, a disaccharidase expressed in the intestinal brush border. Hypomorphic *SI* variants cause recessive congenital sucrase-isomaltase deficiency (CSID) and related gastrointestinal (GI) symptoms. Among children presenting with chronic, idiopathic loose stools, we assessed the prevalence of CSID-associated *SI* variants relative to the general population and the relative GI symptom burden associated with *SI* genotype within the study population.

### Methods

A prospective study conducted at 18 centers enrolled 308 non-Hispanic white children ≤18 years old who were experiencing chronic, idiopathic, loose stools at least once per week for >4 weeks. Data on demographics, GI symptoms, and genotyping for 37 *SI* hypomorphic variants were collected. Race/ethnicity-matched *SI* data from the Exome Aggregation Consortium (ExAC) database was used as the general population reference.

**Funding:** QOL Medical, LLC (https://www.qolmed.com/), funded the entire study. The funders participated in the study design, analyses, and critical review of the manuscript but had no role in data collection, decision to publish, or preparation of the manuscript.

**Competing interests:** Co-authors DC and HS are members of QOL Medical, LLC. QOL Medical LLC markets the product sacrosidase oral solution. This does not alter our adherence to PLOS ONE policies on sharing data and materials.

## Results

Compared with the general population, the cumulative prevalence of hypomorphic *SI* variants was significantly higher in the study population (4.5% vs. 1.3%, $P < .01$; OR = 3.5 [95% CI: 6.1, 2.0]). Within the study population, children with a hypomorphic *SI* variant had a more severe GI symptom burden than those without, including: more frequent episodes of loose stools ($P < .01$), higher overall stool frequency ($P < .01$), looser stool form ($P = .01$) and increased flatulence ($P = .02$).

## Conclusion

Non-Hispanic white children with chronic idiopathic loose stools have a higher prevalence of CSID-associated hypomorphic *SI* variants than the general population. The GI symptom burden was greater among the study subjects with a hypomorphic *SI* variant than those without hypomorphic *SI* variants.

## Introduction

A large number of children around the world experience chronic gastrointestinal (GI) symptoms diagnosed as functional (nonorganic) GI disorders [1]. Functional GI disorders may have several contributing pathophysiologic factors, including carbohydrate malabsorption [2]. The sucrase-isomaltase enzyme, encoded by the *SI* gene, is a predominant member of the disaccharidases responsible for the digestion of dietary carbohydrates in humans [3]. Hypomorphic *SI* gene variants reduce enzyme activity, resulting in congenital sucrase-isomaltase deficiency (CSID) and characteristic GI symptoms. To date, 37 *SI* variants have been identified in diagnosed CSID patients and found to be hypomorphic [4–12]. Patients with CSID experience sucrose malabsorption, leading to colonic osmosis and fermentation, and subsequent osmotic diarrhea and excessive flatulence [4, 13]. Heterozygotes of hypomorphic *SI* variants may also experience GI symptoms. Small case reports have associated hypomorphic *SI* heterozygosity with decreased intestinal sucrase enzymatic activity and characteristic GI symptoms [3, 12, 14]. In two recent studies, the prevalence of heterozygous carriers of hypomorphic *SI* variants was small but significantly greater among adults diagnosed with irritable bowel syndrome (IBS) than in controls, suggestive of an increased IBS susceptibility [15, 16].

A subset of children with functional GI disorders has frequent diarrhea. Currently, the potential contribution of hypomorphic *SI* variants in these children is unknown. Therefore, we had two primary study objectives. The first study objective was to determine the relative prevalence of hypomorphic *SI* variants among children with chronic, idiopathic, loose stools versus the general population. The second study objective in children with chronic, idiopathic, loose stools was to determine the potential impact of hypomorphic *SI* variants on symptom burden.

## Methods

### Study design and subjects

A prospective, 18-center study conducted in the United States enrolled subjects ≤18 years old who presented at a pediatric gastroenterology center with loose stools at least once per week for a minimum of 4 weeks.

The study objectives were i) to determine the prevalence of hypomorphic *SI* variants within the study population versus a genetic database for the general population, and ii) to determine the symptom burden among the study subjects with hypomorphic *SI* variants versus study subjects without hypomorphic *SI* variants.

Exclusion criteria included: identification of any condition(s) or finding(s) that, in the opinion of the investigator, suggested an organic etiology for the subject's GI symptoms; abdominal pain primarily related to constipation; suspected GI infectious disease or other infectious diseases; known GI disease (eg, celiac disease); a history of antibiotic therapy or viral gastroenteritis within the previous 2 weeks; known hepatitis B or C infection or chronic liver disease; cancer or systemic infections; severe neurologic impairment (preventing reporting of symptoms); planned or previous abdominal surgery (eg, bowel resection); severe, uncontrolled systemic diseases; or current use of sacrosidase, an enzyme replacement therapy for CSID.

The pediatric gastroenterology centers in this study were comprised of private practices and academic centers. Each investigator was asked to follow his or her standard clinical practice to ensure an organic etiology was not present. Evaluations took place during a clinic visit. All participating centers received approval from their local institutional review boards (IRB). The institutional review boards included: Baylor College of Medicine IRB, Children's Hospital Los Angeles Committee on Clinical Investigators, Nationwide Children's Hospital IRB, Children's Mercy Hospital Pediatric IRB, Johns Hopkins IRB, The Children's Hospital of Philadelphia Research Institute IRB, The Arnold Palmer Medical Center IRB, The Duke University Health System IRB, Massachusetts General Hospital IRB, Children's Hospital of Wisconsin IRB, Colorado Multiple IRB, The Indiana University IRB, Children's Healthcare of Atlanta IRB, The University of Utah IRB, Children's Hospital & Research Center Oakland IRB, University of Mississippi Medical Center IRB, Columbia University Medical Center IRB, and Sutter Health IRB. All subjects provided assent with a legal guardian providing written informed consent. The study was conducted from May 2013 to July 2015. All authors had access to the study data and approved the final manuscript.

## Genotyping

Four buccal swabs were obtained for DNA extraction; genotyping of the 37 known CSID-associated variants (S1 Table) was completed by a validated capillary electrophoresis assay (SNaP-shot; Laboratory Corporation of America, Research Triangle Park, NC). Intra-assay reproducibility was assessed on 12-sample runs in 3 separate, replicate assays, with samples representing the 37 known hypomorphic variants of the *SI* gene associated with CSID. The reported prevalence of CSID-associated variants included subjects with simple heterozygous, compound heterozygous, and homozygous *SI* variant genotypes. Subjects without hypomorphic *SI* variants did not have any of the 37 analyzed CSID-associated variants on either allele.

The reference for the cumulative prevalence of hypomorphic *SI* variants in the general population was obtained from the Exome Aggregation Consortium (ExAC) database [15]. As the ExAC database data has now been incorporated into the gnomAD project, the evaluated ExAC data is currently available via a legacy link (https://console.cloud.google.com/storage/browser/gnomad-public/legacy). For this study, *SI* sequencing data for 32,550 unrelated non-Hispanic white individuals was retrieved from the ExAC database. The ExAC database provides publicly available genetic data from thousands of unrelated individuals of various races/ethnicities, from aggregated disease-specific and population genetic studies. The ExAC database has been used as a control for genetic studies evaluating a wide range of GI disorders, including adult IBS [15–18].

## Symptom assessment

Subjects were asked to complete a demographic and symptom questionnaire to capture gender, age, race/ethnicity, and episodes of GI symptoms. All GI symptoms (abdominal pain, diarrhea, excessive flatus) were assessed using a study-specific questionnaire, including frequency, duration, and severity of bowel complaints (S2 Table). The GI symptom burden of the ExAC reference population is unknown.

Stool form was assessed using the modified Bristol Stool Form Scale (BSFS; categories 1–5) for children, with higher scores corresponding to looser stools [19]. The symptom questionnaire was generally given to children aged >7 years, with a parent otherwise providing answers.

## Statistical analyses

Results are presented using descriptive statistics, including mean ± standard deviation for continuous data and prevalence and/or percentages for categorical data. The cumulative prevalence of hypomorphic *SI* variants in this study population was compared with a race/ethnicity-matched ExAC prevalence using the Pearson's chi-squared test. Reported *P*-values are one-tailed, and $P < .05$ was considered statistically significant. Increased risk was estimated using the odds ratio (OR) with a 95% confidence interval (CI).

# Results

## Population characteristics

Three hundred and eight non-Hispanic white children with chronic, idiopathic, loose stools were enrolled and assessed. There was a slight predominance of boys (58%). Diarrhea, defined as chronic loose stools, was identified as the primary GI symptom.

## Prevalence of hypomorphic *SI* variants

Among the 308 subjects, 14 had at least 1 hypomorphic *SI* variant, for a prevalence of 4.5%. Study subjects had a statistically significantly higher prevalence of hypomorphic *SI* variants than the race/ethnicity-matched general population (4.5% vs. 1.3%, $P < .01$; OR = 3.5; 95% CI: 6.1, 2.0) (Table 1).

Thirteen of the 14 subjects with an identified hypomorphic *SI* variant were simple heterozygous genotypes (93%), and one subject had a compound heterozygous genotype. Five distinct hypomorphic *SI* variants were identified among these 14 study subjects; 4 of these 5 distinct hypomorphic *SI* variants are the most common *SI* variants found in patients diagnosed with CSID (G1073D, V577G, R1124x and F1745C; Table 2) [12]. There were no statistically significant differences between the prevalence of hypomorphic *SI* variants in male and female subjects (4.5% and 4.7%, respectively).

## Symptom burden associated with hypomorphic *SI* variants

Mean differences in the symptom burden of subjects with a hypomorphic *SI* variant versus subjects without a hypomorphic *SI* variant are reported in Table 3. Compared with the 294 subjects without a hypomorphic *SI* variant, the 14 with a hypomorphic *SI* variant had significantly more frequent GI symptoms, including: more frequent weekly episodes of loose stools ($P < .01$), higher daily overall stool frequency ($P < .01$), looser stool form ($P = .01$) and increased flatulence ($P = .02$). Subjects with a hypomorphic *SI* variant also were younger ($P < .01$).

**Table 1. Relative prevalence of hypomorphic *SI* variants.**

| | Study Population | | ExAC Population | | | | 95% CI | |
|---|---|---|---|---|---|---|---|---|
| | **Number** | **Prevalence** | **Number** | **Prevalence** | **P-Value** | **OR** | **Upper** | **Lower** |
| Wild-type *SI* | 294 | 95.5% | 32,116 | 98.7% | | | | |
| Hypomorphic *SI* variant[a] | 14 | 4.5% | 434 | 1.3% | | | | |
| Total | 308 | | 32,550 | | < .01 | 3.5 | 6.1 | 2.1 |

[a]Includes one compound heterozygote in the study population.

## Discussion

We found hypomorphic *SI* variants among a study population of non-Hispanic white children with chronic, idiopathic, loose stools. The prevalence of these known hypomorphic *SI* variants was significantly higher in this study population compared to a race/ethnicity-matched general population. In addition, study subjects with a hypomorphic *SI* variant had a greater GI symptom burden than study subjects without hypomorphic *SI* variants. These findings add to the growing evidence suggesting heterozygous *SI* hypomorphic variants are associated with the development of CSID-associated GI symptoms.

The most common hypomorphic *SI* variants in our study cohort were also the hypomorphic *SI* variants most commonly identified in other genetic studies of individuals diagnosed with CSID [12]. Although the biochemical and functional effects of several *SI* variants have been well characterized and found to diminish sucrase and isomaltase function [4, 5, 8], the potential effect of a heterozygous genotype of a hypomorphic *SI* variant is still being studied and remains to be fully elucidated. Using either the duodenal disaccharidase enzyme assay or the $^{13}$C breath test to determine the extent of sucrase enzyme activity, family members of patients with CSID (with either presumed or well-documented heterozygosity for a hypomorphic *SI* variant) have been found to have decreased sucrase enzyme activity [14, 20]. Further supporting the potential pathobiological effect of heterozygous genotypes of hypomorphic *SI* variants are recent findings by Henström et al and Garcia-Etxebarria et al, who reported that heterozygous *SI* gene variants may be associated with an increased risk for diagnosis of adult IBS [15, 16]. Nevertheless, further prospective clinical evaluations including identification of heterozygous hypomorphic *SI* variants, functional measurements of sucrase-isomaltase (e.g., enzyme assays), controlled dietary exposures, and basic cellular and molecular based studies are needed to more clearly determine the role of hypomorphic *SI* heterozygous genotypes.

**Table 2. Hypomorphic *SI* variants identified in the study population.**

| *SI* Variants | Rs | Grantham Score[a] | Subjects (N = 308) |
|---|---|---|---|
| G1073D[b] | 121912616 | 94 | 7 |
| V577G[b,c] | 121912615 | 109 | 5[c] |
| F1745C[b] | 79717168 | 205 | 1 |
| R1124x[b] | N/A | N/A | 1 |
| I1378S[c] | 148831941 | 142 | 1[c] |
| Total Unique Subjects | | | 14[c] |

[a]Grantham score is a measure of evolutionary distance in amino acid substitutions, classified by increasing chemical dissimilarity. A higher score reflects a higher likelihood that a substitution will be deleterious based on four general rankings: conservative (0–50), moderately conservative (51–100), moderately radical (101–150), or radical (≥151)

[b]One of the four most common CSID variations

[c]One participant was a compound heterozygote with both a V577G and an I1378S variant.

**Table 3. Symptom burden of study subjects by *SI* genotype.**

|  | Total study population | With Hypo-morphic *SI* Variant | Without Hypo-morphic *SI* Variants | Mean Difference | P-value |
|---|---|---|---|---|---|
| Study subjects | 308 | 14 | 294 |  |  |
| Age (mean yr) | 7.6 | 3.8 | 7.7 | 4.0 | < .01 |
| **Symptom Burden (mean)** |  |  |  |  |  |
| Symptom duration (mo) | 8.8 | 8.6 | 8.9 | 0.3 | NS |
| Diarrhea episode (d/wk) | 5.1 | 6.6 | 5.0 | 1.6 | < .01 |
| Stools (#/d) | 3.4 | 5.3 | 3.3 | 1.9 | < .01 |
| Pediatric BSFS, last diarrheal event | 4.3 | 4.7 | 4.3 | 0.4 | .01 |
| 2Abdominal pain (d/wk) | 3.6 | 3.6 | 3.6 | —— | NS |
| Pain events (#/d) | 1.4 | 1.4 | 1.4 | —— | NS |
| Pain severity (6-point VAS) | 2.4 | 2.4 | 2.4 | —— | NS |
| Gas (d/wk) | 4.9 | 6.3 | 4.9 | 1.4 | .02 |
| Gas events (#/d) | 1.9 | 2.1 | 1.9 | 0.3 | NS |

BSFS, modified Bristol Stool Form Scale for children (categories 1–5, higher scores corresponding to looser stools); NS, not statistically significant; VAS, visual analog scale, 0–5

Identifying the underlying factors contributing to functional GI symptoms is important as this knowledge may lead to more effective therapies. Children with CSID have been shown to benefit from a sucrose-restricted diet and enzyme replacement therapy with sacrosidase taken with meals [21–24]. Gastrointestinal symptoms related to diminished sucrase-isomaltase activity are primarily correlated with factors such as the extent of functional intestinal enzyme activity present and the amount of sucrose and/or starch ingested [4, 13]. It should be noted that in one study, a child with a heterozygous genotype of a hypomorphic *SI* variant, who was a sibling of a CSID-diagnosed subject, was asymptomatic [10]. Future studies in this area may consider assessing both enzyme activity and dietary intake relative to *SI* hypomorphic variants and the associated GI symptom burden.

There are a few limitations to this study. One limitation is that the age-, gender-, and race/ethnicity-matched control population was not actively recruited. However, even though the GI symptoms associated with the database entries are unknown, the ExAC reference database of a significantly large population allowed for a race/ethnicity-matched comparison. Second, the entire *SI* gene was not sequenced in participants. This opens the possibility–however unlikely–that some of the study subjects identified as lacking a hypomorphic *SI* variant had one or more hypomorphic *SI* variants in an uninvestigated portion of the *SI* gene. In addition, some of the study subjects identified as having a simple heterozygous genotype of a hypomorphic *SI* variant may also have a hypomorphic *SI* variant in an uninvestigated portion of the *SI* gene.

There are several strengths in the study. First, this multicenter effort was, to our knowledge, the largest of its kind in children with functional GI symptoms. Second, *SI* genotyping focused on 37 CSID-associated *SI* variants that have been well characterized biochemically as hypomorphic in prior studies. This leads to greater plausibility of the results. Third, the study was conducted in various academic and private practice settings, which will lead to greater generalizability of our findings.

## Conclusion

In conclusion, we found CSID-associated hypomorphic *SI* variants in a study population of non-Hispanic white children with chronic, idopathic, loose stools. These CSID-associated hypomorphic *SI* variants were found to occur at a significantly higher prevalence than that

reported in a race/ethnicity-matched reference population. Subjects with hypomorphic *SI* variants had more GI symptoms of frequent diarrhea and gas, a higher stool frequency, and looser stools compared to those in the study population without hypomorphic *SI* variants.

## Supporting information

**S1 Table. Hypomorphic *SI* Variants Identified in children with chronic gastrointestinal symptoms.** Ala, alanine; Arg, arginine; Asp, aspartate; Cys, cysteine; Gln, glutamine; Glu, glutamate; Gly, glycine; I, isomaltase; Ile, isoleucine; Leu, leucine; N/A, not available; Phe, phenylalanine Pro, proline; S, sucrase; Ser, serine; Thr, threonine; Tyr, tyrosine; Trp, tryptophan; Val, valine. (DOCX)

**S2 Table. Study questionnaire.**
(DOCX)

**S1 Data.**
(XLSX)

## Acknowledgments

We would like to acknowledge all investigators, coordinators, and study site personnel, as well as patients and their families for their participation in this study.

## Author Contributions

**Conceptualization:** Bruno P. Chumpitazi, Derick Cooper.

**Data curation:** Jeffery Lewis, Derick Cooper, Mauro D'Amato, Joel Lim, Sandeep Gupta, Adrian Miranda, Natalie Terry, Devendra Mehta, Ann Scheimann, Molly O'Gorman, Neelesh Tipnis, Yinka Davies, Joel Friedlander, Jaya Punati, Julie Khlevner, Mala Setty, Carlo Di Lorenzo.

**Formal analysis:** Derick Cooper, Mauro D'Amato.

**Funding acquisition:** Derick Cooper.

**Investigation:** Bruno P. Chumpitazi.

**Methodology:** Bruno P. Chumpitazi, Jeffery Lewis, Derick Cooper, Mauro D'Amato, Joel Lim, Sandeep Gupta, Adrian Miranda, Natalie Terry, Devendra Mehta, Ann Scheimann, Molly O'Gorman, Neelesh Tipnis, Yinka Davies, Joel Friedlander, Heather Smith, Jaya Punati, Julie Khlevner, Mala Setty, Carlo Di Lorenzo.

**Supervision:** Derick Cooper.

**Writing – original draft:** Bruno P. Chumpitazi.

**Writing – review & editing:** Bruno P. Chumpitazi, Jeffery Lewis, Derick Cooper, Mauro D'Amato, Joel Lim, Sandeep Gupta, Adrian Miranda, Natalie Terry, Devendra Mehta, Ann Scheimann, Molly O'Gorman, Neelesh Tipnis, Yinka Davies, Joel Friedlander, Heather Smith, Jaya Punati, Julie Khlevner, Mala Setty, Carlo Di Lorenzo.

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
