## [Decision Letter · Decision Letter 0]

31 Jan 2020

PONE-D-19-34036

Hypomorphic SI genetic variants are associated with childhood chronic loose stools

PLOS ONE

Dear Dr. Chumpitazi,

Thank you for submitting your manuscript to PLOS ONE. After careful consideration, we feel that it has merit but does not fully meet PLOS ONE’s publication criteria as it currently stands. Therefore, we invite you to submit a revised version of the manuscript that addresses the points raised during the review process.

The Authors should implement the discussion following the suggestions of the Reviewer.

We would appreciate receiving your revised manuscript by Mar 16 2020 11:59PM. To enhance the reproducibility of your results, we recommend that if applicable you deposit your laboratory protocols in protocols.io, where a protocol can be assigned its own identifier (DOI) such that it can be cited independently in the future. For instructions see: http://journals.plos.org/plosone/s/submission-guidelines#loc-laboratory-protocols

We look forward to receiving your revised manuscript.

Kind regards,

Adriana Calderaro

Academic Editor

PLOS ONE

Journal Requirements:

2. Please provide additional details regarding participant consent. In the ethics statement in the Methods and online submission information, please ensure that you have specified whether consent was suitably informed. Since your study included minors under age 18, please also state whether you obtained consent from parents or guardians.

3. Thank you for your ethics statement : "All participating centers received approval from their local institutional review boards. Baylor College of Medicine IRB protocol number was H-32378. All subjects provided written consent."

"The entire study was funded by QOL Medical, LLC (https://www.qolmed.com/). The funders participated in the study design, analyses, and in preparation of the manuscript. "

6. Thank you for stating the following in the Competing Interests section:

"Co-authors DC and HS are members of QOL Medical, LLC."

Reviewers' comments:

Reviewer's Responses to Questions

**Comments to the Author**

1. Is the manuscript technically sound, and do the data support the conclusions?

Reviewer #1: Yes

Reviewer #2: Yes

2. Has the statistical analysis been performed appropriately and rigorously? 

Reviewer #1: Yes

Reviewer #2: Yes

3. Have the authors made all data underlying the findings in their manuscript fully available?

Reviewer #1: Yes

Reviewer #2: Yes

4. Is the manuscript presented in an intelligible fashion and written in standard English?

Reviewer #1: Yes

Reviewer #2: Yes

5. Review Comments to the Author

Reviewer #1: The manuscript compiles the results of a large multicenter study on the relation between functional GI symptoms in children and 37 variants of the sucrase-isomaltase gene that have been previously characterized at the biochemical and cellular levels in congenital sucrase-isomaltase deficiency (CSID). The study demonstrates a significantly higher prevalence of hypomorphic SI variants in the cases studied than in the general population.

The study supports recent views that heterozygous genotype of hypomorphic SI may be associated with an increased risk of irritable bowel syndrome. The study is properly conducted and impressive in terms of the number of cases studied. The statistical analyses are convincing and the conclusions are sound. The manuscript is clearly written (only lines 95/96 should be rewritten).

I fully agree with the conclusions, have no points of criticism and recommend the acceptance of the manuscript in its present form.

Reviewer #2: The manuscript by Chumpitazi et al assesses the prevalence of variants in the sucrase-isomaltase gene among a cohort of children with chronic diarrhea. The authors find that confirmed or predicted loss of function mutations are more prevalent in the studied population versus a the EXAC reference genomic database. Furthermore, they show that patients with mutations have more diarrhea symptoms than the rest of the patient cohort. The study suggests a potentially interesting possible association between mild chronic diarrhea and SI function in young children. The strengths of the paper are that the patient cohort is taken from multiple different centers / sites and sample genotyping was done in a uniform manner at a central site. The statistics and results, while limited in scope are appropriately done.

1. The major limitation in understanding the significance of these findings, as briefly but insufficiently discussed, is that the functional effect of heterozygosity in the various loss-of-function mutations, and especially the common variants found in these patients remain poorly described and essentially unknown. The authors cite a single case report and a study on 13C breath testing in functional GI disorders (not specifically with known SI genotype), neither of which is sufficient to address the question of SI heterozygosity and enzyme activity. To assess the functional significance requires prospective clinical studies with functional measurements (enzyme/breath testing) correlated to genotype as well as basic cell-based studies assessing heterozygous vs homozygous mutations and function / localization. This should be more clearly part of the discussion and the major limitations of the existing literature (studies of symptom correlation rather than objective enzyme activity, few studies, no cellular studies etc) more straightforwardly stated.

2. In the methods the reference database link should be updated to either Gnomad or the legacy EXAC only data. There should be brief sentence explaining this update.

6. PLOS authors have the option to publish the peer review history of their article (what does this mean?). If published, this will include your full peer review and any attached files.

Reviewer #1: No

Reviewer #2: No

---

## [Author Response · Author response to Decision Letter 0]

3 Apr 2020

Reviewer #1: The manuscript compiles the results of a large multicenter study on the relation between functional GI symptoms in children and 37 variants of the sucrase-isomaltase gene that have been previously characterized at the biochemical and cellular levels in congenital sucrase-isomaltase deficiency (CSID). The study demonstrates a significantly higher prevalence of hypomorphic SI variants in the cases studied than in the general population.

The study supports recent views that heterozygous genotype of hypomorphic SI may be associated with an increased risk of irritable bowel syndrome. The study is properly conducted and impressive in terms of the number of cases studied. The statistical analyses are convincing and the conclusions are sound. The manuscript is clearly written (only lines 95/96 should be rewritten).

I fully agree with the conclusions, have no points of criticism and recommend the acceptance of the manuscript in its present form.

We thank the Reviewer for his/her highly supportive comments of the manuscript. We have rewritten lines 95-96 as requested.

Reviewer #2: The manuscript by Chumpitazi et al assesses the prevalence of variants in the sucrase-isomaltase gene among a cohort of children with chronic diarrhea. The authors find that confirmed or predicted loss of function mutations are more prevalent in the studied population versus a the EXAC reference genomic database. Furthermore, they show that patients with mutations have more diarrhea symptoms than the rest of the patient cohort. The study suggests a potentially interesting possible association between mild chronic diarrhea and SI function in young children. The strengths of the paper are that the patient cohort is taken from multiple different centers / sites and sample genotyping was done in a uniform manner at a central site. The statistics and results, while limited in scope are appropriately done.

1. The major limitation in understanding the significance of these findings, as briefly but insufficiently discussed, is that the functional effect of heterozygosity in the various loss-of-function mutations, and especially the common variants found in these patients remain poorly described and essentially unknown. The authors cite a single case report and a study on 13C breath testing in functional GI disorders (not specifically with known SI genotype), neither of which is sufficient to address the question of SI heterozygosity and enzyme activity. To assess the functional significance requires prospective clinical studies with functional measurements (enzyme/breath testing) correlated to genotype as well as basic cell-based studies assessing heterozygous vs homozygous mutations and function / localization. This should be more clearly part of the discussion and the major limitations of the existing literature (studies of symptom correlation rather than objective enzyme activity, few studies, no cellular studies etc) more straightforwardly stated.

We thank the Reviewer for his/her supportive comments and critiques of the manuscript. We agree that the functional effect of heterozygosity for the identified mutations needs to be further elucidated. We have edited the Discussion within the manuscript to more clearly delineate the need for further investigation.

2. In the methods the reference database link should be updated to either Gnomad or the legacy EXAC only data. There should be brief sentence explaining this update.

We have updated the reference database link and provided an explanation for the update

---

## [Decision Letter · Decision Letter 1]

3 Apr 2020

Hypomorphic SI genetic variants are associated with childhood chronic loose stools

PONE-D-19-34036R1

Dear Dr. Chumpitazi,

We are pleased to inform you that your manuscript has been judged scientifically suitable for publication and will be formally accepted for publication once it complies with all outstanding technical requirements.

With kind regards,

Adriana Calderaro

Academic Editor

PLOS ONE

Additional Editor Comments (optional):

Reviewers' comments:

Reviewer's Responses to Questions

**Comments to the Author**

1. If the authors have adequately addressed your comments raised in a previous round of review and you feel that this manuscript is now acceptable for publication, you may indicate that here to bypass the “Comments to the Author” section, enter your conflict of interest statement in the “Confidential to Editor” section, and submit your "Accept" recommendation.

Reviewer #1: All comments have been addressed

Reviewer #2: All comments have been addressed

2. Is the manuscript technically sound, and do the data support the conclusions?

Reviewer #1: Yes

Reviewer #2: Yes

3. Has the statistical analysis been performed appropriately and rigorously? 

Reviewer #1: Yes

Reviewer #2: Yes

4. Have the authors made all data underlying the findings in their manuscript fully available?

Reviewer #1: Yes

Reviewer #2: Yes

5. Is the manuscript presented in an intelligible fashion and written in standard English?

Reviewer #1: Yes

Reviewer #2: Yes

6. Review Comments to the Author

Reviewer #1: My initial review recommended acceptance of the manuscript as it stands and I have just asked for minor edits. These edits have been made in the revised version.

Reviewer #2: (No Response)

7. PLOS authors have the option to publish the peer review history of their article (what does this mean?). If published, this will include your full peer review and any attached files.

Reviewer #1: No

Reviewer #2: No

---

## [Editor Report · Acceptance letter]

8 May 2020

PONE-D-19-34036R1 

Hypomorphic *SI* genetic variants are associated with childhood chronic loose stools 

Dear Dr. Chumpitazi:

I am pleased to inform you that your manuscript has been deemed suitable for publication in PLOS ONE. Congratulations! Your manuscript is now with our production department. 

With kind regards,

on behalf of

MD, PhD, Associate Professor Adriana Calderaro 

Academic Editor

PLOS ONE